# Transparent proton transport through a two-dimensional nanomesh material

Jiyu Xu [1,2,3,6], Hongyu Jiang[1,2,3,6], Yutian Shen[1,2,3], Xin-Zheng Li[4,5], E.G. Wang[1,3,4,5] & Sheng Meng [1,2,3,5]

Molecular sieving is of great importance to proton exchange in fuel cells, water desalination, and gas separation. Two-dimensional crystals emerge as superior materials showing desirable molecular permeability and selectivity. Here we demonstrate that a graphdiyne membrane, an experimentally fabricated member in the graphyne family, shows superior proton conductivity and perfect selectivity thanks to its intrinsic nanomesh structure. The transmembrane hydrogen bonds across graphdiyne serve as ideal channels for proton transport in Grotthuss mechanism. The free energy barrier for proton transfer across graphdiyne is ~2.4 kJ mol$^{-1}$, nearly identical to that in bulk water (2.1 kJ mol$^{-1}$), enabling "transparent" proton transport at room temperature. This results in a proton conductivity of 0.6 S cm$^{-1}$ for graphdiyne, four orders of magnitude greater than graphene. Considering its ultimate pore size of 0.55 nm, graphdiyne membrane blocks soluble fuel molecules and exhibits superior proton selectivity. These advantages endow graphdiyne a great potential as proton exchange material.

[1] Beijing National Laboratory for Condensed Matter Physics and Institute of Physics, Chinese Academy of Sciences, Beijing 100190, People's Republic of China. [2] School of Physical Sciences, University of Chinese Academy of Sciences, Beijing 100049, People's Republic of China. [3] Songshan Lake Materials Laboratory and School of Physics, Liaoning University, Dongguan, Guangdong 523808, People's Republic of China. [4] State Key Laboratory for Mesoscopic Physics and School of Physics, Peking University, Beijing 100871, People's Republic of China. [5] Collaborative Innovation Center of Quantum Matter, Beijing 100871, People's Republic of China. [6] These authors contributed equally: Jiyu Xu, Hongyu Jiang. Correspondence and requests for materials should be addressed to E.G.W. (email: egwang@iphy.ac.cn) or to S.M. (email: smeng@iphy.ac.cn)

Molecular sieving is of great importance to proton exchange membranes (PEMs) for fuel cells (FCs)[1,2], water desalination[3], and gas separation[4]. Due to the ultrathin film thickness and high mechanical strength, two-dimensional (2D) materials are promising for molecular sieving and exhibit desirable molecular permeability and selectivity[4–8] that is dependent on the natural[9] or fabricated[10–12] pores in membrane planes. Recently, proton conductivity has been realized in 2D materials, for example, graphene[13–15], graphene oxides[16], and hexagonal boron nitride (h-BN)[14,15,17]. Theoretical works demonstrate that the energy barriers are very large (>1.5 eV) for proton, the lightest nuclei, to transport through non-defective graphene[18–22], whether in chemisorption or physisorption processes. The emergence of transport channels accounts for improved proton conductivity[21–23]. For instance, defects such as Stone–Wales defect are known to reduce the energy barrier of proton transfer through graphene in vacuum from 1.5 to 0.9 eV[21]. However, the proton conductivity from naturally occurring defects is low due to their relatively small density on graphene membrane[14].

The density of nanopores for proton transport (PT) can be greatly enhanced by oxidation of graphene, which simultaneously gives rise to functional groups, for example, hydroxyl and epoxy moieties, decorating the membranes. Consequently, graphene oxides exhibit nice proton conductivity across membrane in aqueous condition, via interactions between water and functional groups on the periphery of nanopores[16]. However, it is hard to control the oxidation procedure to achieve uniform pore sizes and decoration. Large pinholes can be generated, which greatly deteriorates proton selectivity, a strongly desired property in realistic applications. For example, proton selectivity over methanol molecules is required for PEM applications in direct methanol FCs. There is a balance required between proton conductivity and selectivity, the two competing factors strongly dependent on the size of transport channels.

For PEMs in FCs, the optimal size of channel is supposed to be between proton and soluble fuel molecules to achieve both good conductivity and selectivity. Meanwhile the membranes also need to exhibit large channel density and great channel uniformity. However, besides oxidation, other technologies for pore drilling, for example, ion bombardment[11,12], also fail to generate such delicate channels. Fortunately, the intrinsic 2D nanomesh materials exhibit periodically distributed nanopores with greatest pore density and uniformity, and thus show a great potential as molecular sieving materials[9,24]. Among them, graphdiyne[25], also called graphyne-2, as the experimentally fabricated 2D nanomesh membrane in the graphyne family[26–29], exhibits excellent mechanical, physical, and electrochemical properties[30–36]. Graphdiyne shows an ultimate pore size of 0.55 nm (van der Waals pore size of 0.06 nm$^2$) and the unprecedented nanopore density of $2.5 \times 10^{18}$ m$^{-2}$. Activated transport is found for $H_2$, CO, $CH_4$, and $H_2O$ molecules due to its ultimate pore size[34–36]. However, despite the optimal pore size and superior pore uniformity, the proton conductivity and selectivity in aqueous solutions have not been tested for graphdiyne.

In this work, we demonstrate using extensive ab initio molecular dynamics (AIMD) simulations and density functional theory (DFT) calculations that graphdiyne membrane exhibits superior proton conductivity and selectivity in aqueous solutions at room temperature (Fig. 1). Trans-membrane (TM) hydrogen bonds (HBs) across graphdiyne membrane serve as an ideal channel for proton transfer in a Grotthuss mechanism. The free energy barrier for proton transfer across membrane is found to be ~2.4 kJ mol$^{-1}$ at room temperature, nearly identical to that in bulk water (~2.1 kJ mol$^{-1}$). Thus, protons can freely diffuse through graphdiyne membrane triggered by thermal fluctuations. The corresponding

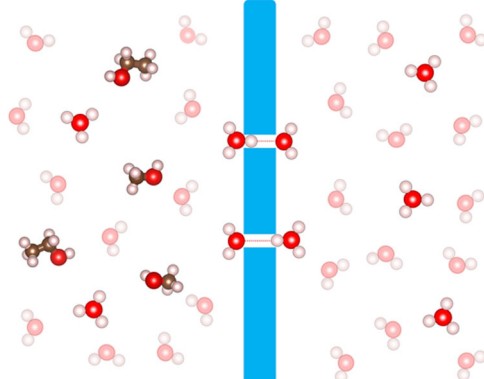

**Fig. 1** A schematic illustration for proton conductivity and selectivity of graphdiyne membrane. Color code: C, gray; H, white; O, red; graphdiyne membrane, blue; hydrogen bond, red dot line

proton conductivity of graphdiyne membrane is estimated to be 0.6 S cm$^{-1}$, which is four orders of magnitude greater than graphene and one order of magnitude greater than the commercial PEM material Nafion. The rate of PT can be further enhanced when nuclear quantum effects (NQEs) have been taken into account. In addition, the optimal nanopore size gives rise to superior proton selectivity, while transport of soluble fuel molecules is fully blocked. Therefore, graphdiyne shows a great potential as PEM materials in FCs, sensors, and other applications.

## Results

**Proton diffusion in the vicinity of membrane.** To verify proton conductivity of graphdiyne membrane, we performed two sets of long-time equilibrium AIMD simulations of proton diffusion at water–graphdiyne interfaces. In the simulations, 32 water molecules are equally distributed on both sides of the graphdiyne membrane, and thus the thicknesses of the water layers on the two sides are both about 8 Å. The two sets of equilibrium AIMD simulations were performed with two different initial positions of $H_3O^+$ complexes. (The definition of proton position is shown in Supplementary Fig. 1 and Supplementary Note 1.) In the first group, a $H_3O^+$ complex is initially located in the bulk water layers below the membrane, while a $H_3O^+$ complex is put directly beneath the nanopore of the graphdiyne membrane in the second group. After optimization, total AIMD simulations of 120 ps were performed at 300 K to reveal the PT behavior at the interface.

Surprisingly, a proton diffuses through graphdiyne membrane from the water layers below the membrane to that above the membrane in Traj_1 of Fig. 2a. The TM PT phenomenon occurs under unbiased conditions, which indicates that thermal fluctuations trigger the TM PT and graphdiyne membrane exhibits weak hindrance for TM PT. However, the proton complexes are still located in the water layers below the membrane in other trajectories Traj_2, Traj_3, and Traj_4. We can get more insight of interfacial proton diffusion in the second set of simulations due to the close initial contact of proton and graphdiyne membrane as discussed below. As shown in Fig. 2b, in the second set of simulations proton frequently transfers across the graphdiyne atomic plane back and forth in the first 3 ps, verifying the superior efficiency of TM PT. Then, the protons diffuse through graphdiyne membrane to the water layers above the membrane in Traj_1 and Traj_2, and off membrane to the water layers below the membrane in Traj_5 and Traj_6, and they are still close to the graphdiyne membrane in Traj_3 and Traj_4. The diverse directions lead to a nearly uniform distribution of protons in the water layers. Meanwhile, the picosecond timescale for proton transfer away from membrane indicates that graphdiyne

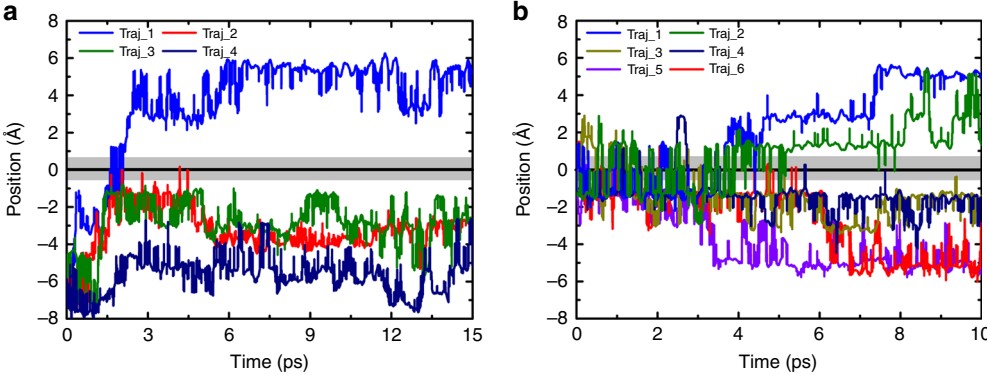

**Fig. 2** Diffusion of protons at water–graphdiyne interfaces. The trajectories of protons with proton initially located **a** in the bulk water layer below graphdiyne membrane and **b** right beneath the nanopore of graphdiyne membrane

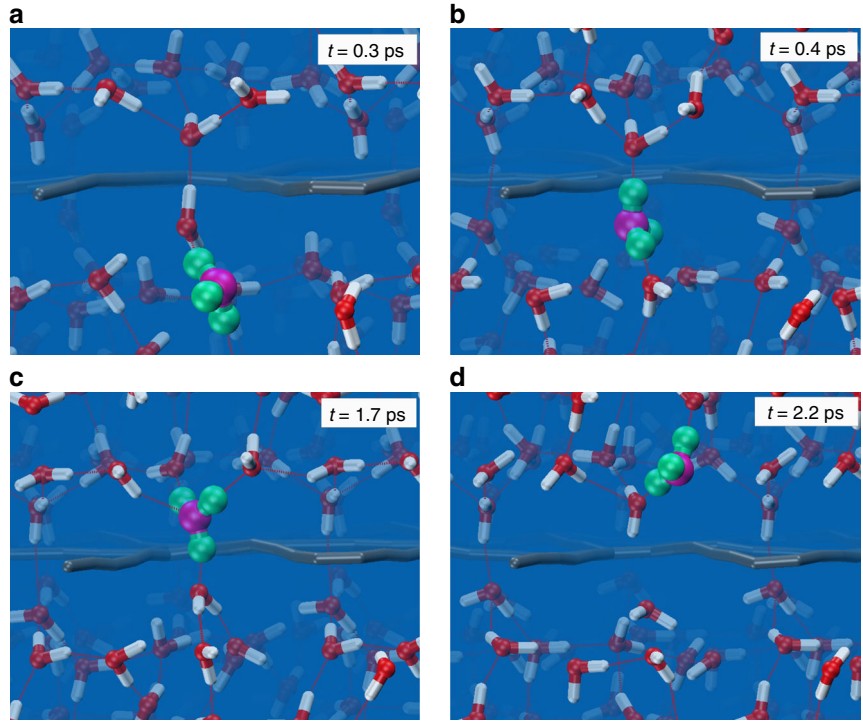

**Fig. 3** The process of trans-membrane proton transport. **a–d** The chronological snapshots for proton transport across graphdiyne membrane via the Grotthuss mechanism in Traj_1 of Fig. 2a. Color code: C, gray; H, white; O, red; H in $H_3O^+$ complex, green; O in $H_3O^+$ complex, magenta

membrane does not block proton diffusion at all in aqueous solution at ambient conditions.

To gain deeper insight for TM PT, we track the Traj_1 in Fig. 2a. Figure 3a–d show the snapshots of TM PT process chronologically. Proton transfers sequentially from the lower water molecule in Fig. 3a to the upper water molecules in Fig. 3b–d via HBs in the Grotthuss mechanism, with four oxygen atoms acting as intermediate hopping sites. In the above process, a TM HB emerges and connects the lower and upper water layers, thus constituting a consecutive channel for proton transfer across the graphdiyne membrane. We note that proper dipole orientation of the water molecule in the transport channel is necessary for TM PT. This characteristic interfacial water structure is indeed resulted from the porous morphology of graphdiyne membrane[36].

**Interfacial water structure**. Classical molecular dynamics simulations (Supplementary Method 1) confirm the characteristic interfacial structure of water. As shown in Fig. 4a, a peak (region I) in the water density profile emerges besides the conventional water layer[37] (region II). The peak in region I corresponds to water molecules located right above each nanopore in Fig. 3b–c. We note that almost every nanopore in the graphdiyne membrane is occupied by water molecules. The conventional water layer in region II corresponds to water molecules at the surface of graphdiyne (similar to water molecules in the case marked in Fig. 3a, d). We define water molecules at region I as active water, water molecules in region II as interfacial water, and others as bulk water. Importantly, although separated by graphdiyne membrane, the two active water molecules from both sides of a nanopore form a TM HB at the probability of 70% even at ambient conditions. Besides this TM HB, the active water molecule forms 2 or 3 additional HBs with interfacial water molecules in region II. No HB forms between two adjacent active water molecules on the same side due to the large separation of 5.5 Å between adjacent nanopore centers. We note that the adequate

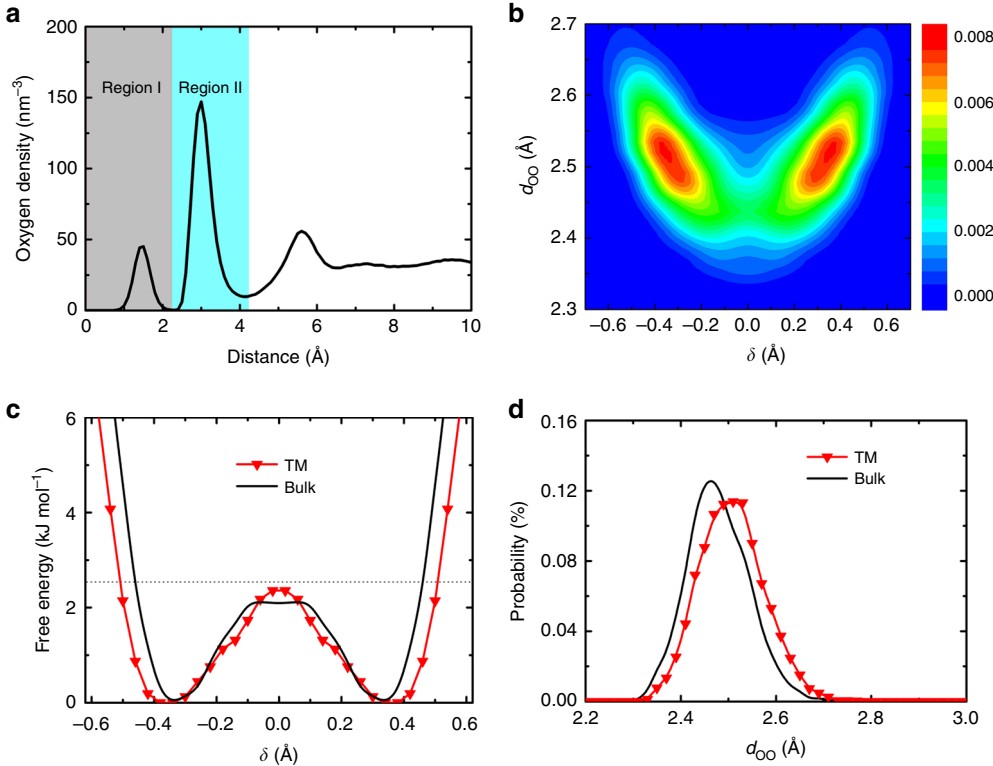

**Fig. 4** Transparent proton transport via the characteristic interfacial water structure. **a** Oxygen density distribution along surface normal of graphdiyne membrane. **b** Probability distribution of the excess proton as a function of $\delta$ and $d_{OO}$ in trans-membrane (TM) step (i) sampled in all 10 equilibrium simulations. **c** The free energy profile for the excess proton in TM step (i) and in bulk water. The thermal energy $k_B T$ at 300 K is marked as the horizontal dashed line. **d** Distribution of $d_{OO}$ for the excess proton in TM step (i) and in bulk water

pore size and the hydrophobic pore rim is the key to the formation of stable TM HBs. TM HBs cannot form across graphyne-1 and graphene membranes.

**TM free energy barrier**. To characterize proton transfer process, we define transfer coordinate $\delta$ of each proton as the distance difference between the proton and its two nearest oxygen atoms (O1 and O2), $\delta = d_{HO1} - d_{HO2}$. Thus, the excess proton is the proton with the smallest transfer coordinate $\delta$. The species of proton complex are differentiated on the basis of transfer coordinate $\delta$ (Supplementary Fig. 1 and Supplementary Note 1). Associated with $\delta$, $d_{OO}$ is defined as the distance between the O1 and O2 atom. The shape of the potential energy surface for proton transfer as a function of $\delta$ is closely related to the value of $d_{OO}$, with smaller $d_{OO}$ normally gives a smaller proton transfer barrier.

Considering the unique interfacial water structures, we divide the PT process across the membrane from region I′ (active layer on one side of the membrane) to region II (interfacial layer on the other side of the membrane) into three sequential steps: (i) TM transfer (region I′ ↔ I, I denotes the region on the other side of the membrane), (ii) switch of excess proton on the same oxygen atom located in region I; and (iii) active-to-interfacial (region I ↔ II) transfer.

We focus on the TM step (i) firstly. Figure 4b shows the probability distribution of the excess proton as a function of $\delta$ and $d_{OO}$ in TM step (i) sampled in equilibrium simulations above. The double-peak structure indicates that proton is mainly $H_3O^+$ complex shown in Fig. 3b–c, and the reduced probability at $\delta = 0$ corresponds to the transition state of $H_5O_2^+$ complex. We extract the free energy of proton transfer as: $\Delta F = -k_B T \ln P$, where $P$ is

the probability as a function of proton transfer coordinate $\delta$, $T$ is temperature, and $k_B$ is the Boltzmann constant. As shown in Fig. 4c, the free energy barrier for proton transfer across graphdiyne membrane is smaller than $k_B T$ at 300 K, indicating that thermal fluctuations are sufficient to drive the TM PT. The barrier (2.4 kJ mol⁻¹) for PT across the graphdiyne membrane is slightly greater than that (2.1 kJ mol⁻¹) in bulk water (Supplementary Note 2). Besides, the minima on the free energy profile of TM PT is 0.04 Å farther away from $\delta = 0$. Both are attributed to that the $d_{OO}$ in TM HB is 0.04 Å greater than that in bulk water as shown in Fig. 4d. Despite the small difference, the same energy profiles demonstrate that the TM PT is easily driven by thermal fluctuations, in the same manner to that in bulk water. Although proton diffusion involves breaking of OH bonds, the excess proton could not bond to graphdiyne membrane (Supplementary Fig. 2), which is attributed to the hydrophobic effects of the inert and neutral pore rim. Besides, the TM $H_5O_2^+$ complex is the most stable proton complex in vacuum, shown in Supplementary Fig. 3 and Supplementary Note 3. Thus, the inert nanopore on graphdiyne membrane only serves as a spatial constraint for TM proton transfer.

The free energy profile in the active-to-interfacial (region I ↔ II) proton transfer step (iii) is asymmetric as shown in Supplementary Fig. 4, and proton slightly prefers to bond to the active water. However, the barrier for proton transfer off the active water molecule is only 0.7 kJ mol⁻¹ larger than $k_B T$, and protons could diffuse into the bulk water at a picosecond timescale in MD simulations. The step (i) and step (iii) can be connected via the switch of the excess proton on the oxygen atom at region I via step (ii). The excess proton can be one of the three protons bound to the single oxygen, where one proton corresponds to TM proton transfer and two protons correspond

to active-to-interfacial proton transfer, shown in Fig. 3b. We count the two kinds of excess protons exhibiting proton transfer. Statistical result shows that the ratio between the two kinds of excess protons is 1:2. Besides, the excess proton switches quickly between the two kinds of protons at a timescale of 10 fs, implying that the proton diffusion proceeds forward and backward freely at the interface. Therefore, via the three steps discussed above, proton freely diffuses across graphdiyne membrane as in bulk water, and graphdiyne membrane is almost "transparent" for proton diffusion.

**PT driven by electric field.** Besides the equilibrium statistics, a set of ten AIMD simulations under electric field were also performed to simulate PT phenomena across graphdiyne membrane under non-equilibrium conditions. More specifically, two protons were placed on the same side away from graphdiyne membrane with an electric field of 0.1 V Å$^{-1}$ to accelerate PT in these simulations (Supplementary Fig. 5). The strong acidity (~4 M) and large electric field are included to introduce a directional transport of proton and to get around of the unprecedented heavy computation cost. Considering the nature of graphdiyne membrane, we take the distance between the two interfacial water layers as the effective thickness of graphdiyne membrane in the series circuit of water–membrane–water system. Then, we define the residence time $\tau$ as the time interval between the first excess proton reaching the lower interfacial water in Fig. 3a and that reaching the upper interfacial water in Fig. 3d, involved in the three steps above, to quantify the transport rate of graphdiyne membrane. The result shows that the average residence time $\tau$ is 0.47 ps. The corresponding proton conductivity is 8.7 S cm$^{-1}$ for a single nanopore. Assuming that proton is equally distributed in solution, we estimate the proton conductivity of graphdiyne membrane is 0.6 S cm$^{-1}$, which is one order of magnitude greater than the commercial Nafion[38]. Besides, compared with other 2D materials, the areal proton conductivities of graphdiyne is four orders of magnitude greater than graphene at ~500 K[14], and two orders of magnitude greater than graphene oxide[16].

We extract the free energy profile for PT sampled during the residence time, which is shown in Fig. 5b. Compared to the classical barrier without electric field in Fig. 4c, we see that the barrier is reduced by 1 kJ mol$^{-1}$ due to the acceleration effect of electric field, which is consistent with the fast transport of excess proton through graphdiyne membrane. We can estimate the barrier reduction by electric field schematically with $\Delta E \sim F dq$, where $F$ represents the electric field that takes a value of 0.1 V Å$^{-1}$ here, $d$ is the displacement of excess proton during proton

transfer taking a value of 0.34 Å (the distance from energy minimum to $\delta = 0$), and $q$ is the effective charge of excess proton in hydronium molecule (~0.4$e$). The estimated barrier reduction of 1.3 kJ mol$^{-1}$ is consistent with the statistical result obtained from MD simulations.

Another issue to be considered is the NQEs, as proton is the lightest nucleus and PT occurs in a confined system. This is carried out by performing ab initio path-integral molecular dynamics (PIMD) simulations under the same electric field and comparing the results with that of the AIMD simulations. Figure 5a shows a typical quantum configuration of TM H$_5$O$_2^+$ complex in transport processes. It is obvious that the zero-point energy effects lead to the swelling of proton nuclei. Besides this, the free energy profile was calculated using the same equation as the classical one. From Fig. 5b, we see that this barrier is further decreased by 0.8 kJ mol$^{-1}$ upon including the NQEs. Besides, the minima on the free energy profile of quantum proton are 0.1 Å closer to $\delta = 0$. Both indicate that including the NQEs greatly enhances the efficiency of proton transfer. Considering the small energy difference of 0.3 kJ mol$^{-1}$ between PT in bulk water and TM PT using classical nuclei in Fig. 4c, the graphdiyne membrane is more transparent upon including the NQEs.

**Feasibility as PEMs.** To compare with other 2D materials facilely, we perform climbing image nudged elastic band (CI-NEB) calculations to get the energy barriers for proton transfer through a variety of 2D materials (Supplementary Fig. 6). Two water molecules are located on each side of membrane to include the aqueous effects. Considering the fact that the proton complex H$_5$O$_2^+$ is stable across graphdiyne membrane, we fix the distance of two oxygen atoms to 2.8 Å to get an estimate of barrier maximum in simulations of graphdiyne membrane. The distance of 2.8 Å is large enough to accommodate proton transfer events in AIMD simulations (Fig. 4d). As shown in Fig. 6, the energy barriers for non-defective graphene (4.04 eV) and h-BN (3.48 eV) are too large for proton to transfer. With the emergence of nanopores in β-boron and graphyne-1, the barrier greatly decreases to ~1 eV. No TM HB can be formed across these membranes above. While, via the preformed TM HB, an ultralow barrier of 0.15 eV is obtained for proton transfer across 0.55-nm-diameter nanopore on graphdiyne. Furthermore, in realistic aqueous solutions, this barrier decreases owing to fluctuations in $d_{OO}$, consistent with the free energy barrier of 2.4 kJ mol$^{-1}$ (0.025 eV) calculated above. Comparing to previous results, PT across graphdiyne membrane exhibits no extra requirements, for example, decoration of nanopores[13], hydrogenation, and

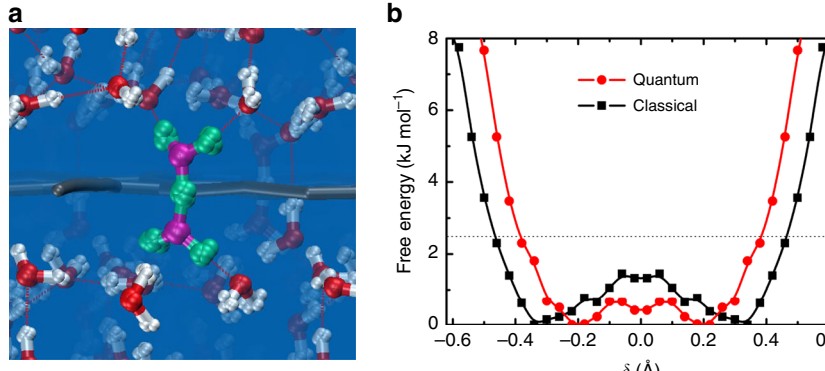

**Fig. 5** Trans-membrane (TM) proton transport driven by electric field. **a** Typical quantum configuration of TM H$_5$O$_2^+$ complex in path-integral molecular dynamics (PIMD) simulations. **b** The free energy profile of the excess proton during proton transport processes from classical and quantum nuclei simulations under electric field, respectively. The profiles are symmetrized for a more accurate quantification of the barrier. The thermal energy $k_B T$ at 300 K is denoted by the horizontal dashed line

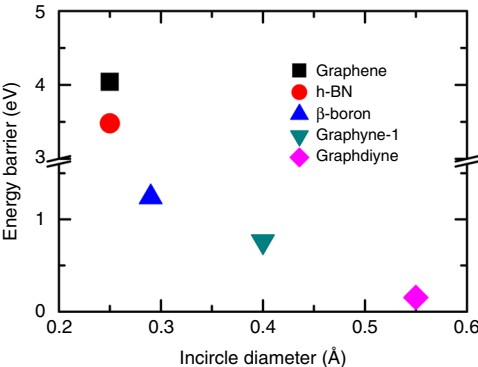

**Fig. 6** Energy barriers for proton transport across different two-dimensional (2D) membranes

**Table 1 Energy barriers for solutes in aqueous solution passing through nanopore in graphdiyne membrane**

| Species | Proton | CH$_3$OH | CH$_3$CH$_2$OH | Na$^+$ | Cl$^-$ |
|---|---|---|---|---|---|
| Energy barrier (eV) | 0.15 | 1.77 | 3.66 | 1.09 | 1.4 |

distortion of membranes[39], to attain a low energy barrier. Besides, the great nanopore density of $2.5 \times 10^{18}\,\mathrm{m}^{-2}$ and uniformity of nanopores in graphdiyne membrane greatly enhance the PT efficiency in aqueous solutions.

The TM HBs across graphdiyne offer a fast channel for proton transfer in the Grotthuss mechanism, which accounts for the proton transparency property of graphdiyne membrane. The proton transparency property is further demonstrated with nearly the same energy barrier profiles of proton transfer in free and TM H$_3$O$^+$–H$_2$O complexes (Supplementary Fig. 7c). Besides, benchmark calculations with the hybrid functional (Supplementary Note 4) confirm the proton transparency property of graphdiyne membrane (Supplementary Figs. 7d, 8, and 9) and the existence of TM HBs (Supplementary Fig. 10). We note that the TM structures can be regarded as the shortest one-dimensional water wires as those in 0.8-nm-diameter carbon nanotubes, where superior PT rate is also obtained[40]. The hydrophobic pore rim facilitate the formation of transport channel of this short one-dimensional water wire in graphdiyne nanopore, in the same way as that in the hydrophobic inner wall of carbon nanotubes.

We note that H$_3$O$^+$ complex directly penetrating the nanopore in graphdiyne membrane could offer another transport channel for TM PT. The CI-NEB calculations (Supplementary Fig. 6) show an energy barrier of 0.55 eV. However, the relatively large barrier of 0.55 eV results in a small contribution to TM PT compared with the ultralow barrier of 0.025 eV in the Grotthuss mechanism. Meanwhile, in PEM applications, the driven force is electrical field or concentration gradient. It is distinctly different from water transport where the energy barrier can be modulated by large hydrostatic pressure[36]. We conclude that this transport channel plays a secondary role in TM PT at ambient conditions, which albeit enhances the efficiency of TM PT through graphdiyne membrane.

To verify the proton selectivity of graphdiyne membrane, we calculate energy barriers of soluble fuel molecules and ions passing through the graphdiyne membrane with scanning-path method (Supplementary Fig. 11). As shown in Table 1, graphdiyne membrane shows great barriers for CH$_3$OH (1.77 eV) and CH$_3$CH$_2$OH (3.66 eV), and impermeable for both soluble molecules due to the great molecular volume. Furthermore, graphdiyne membrane also shows great barriers for Na$^+$ (1.09 eV) and Cl$^-$ (1.4 eV) due to the dehydration effects of ions, thus the transport of ions is blocked[36]. Thus, graphdiyne membrane shows a superior proton selectivity in aqueous solution at ambient conditions.

The superior proton conductivity of graphdiyne membrane is attained at room temperature, which is completely different from proton conductivity of graphene promised at elevated temperature[14]. The optimal pore size not only gives rise to the nice proton conductivity, but allows the excellent separation efficiency of proton from soluble fuel molecules and ions. It nicely addresses the severe issues of high fuel permeability and rigorous working conditions, which limit the efficiency of commercial PEM Nafion. Besides, the ultrathin graphdiyne membrane greatly reduces the electrical resistance of PEM in the circuit. Considering the superior proton conductivity and selectivity, as well as perfect mechanical and chemical stability, graphdiyne shows a great potential as the next generation of PEM materials.

## Discussion

In conclusion, we demonstrate that graphdiyne membranes with a dense, uniform 2D nanomesh structure (nanopore density of $2.5 \times 10^{18}\,\mathrm{m}^{-2}$) show superior proton conductivity and selectivity. Protons diffuse through graphdiyne membrane in the Grotthuss mechanism via TM HBs. The barrier for PT across membrane is comparable with that in bulk water at ambient conditions; thus, thermal fluctuation can effectively trigger the TM PT phenomena. The calculated proton conductivity of graphdiyne membrane is 0.6 S cm$^{-1}$, four orders of magnitude greater than graphene and one order of magnitude greater than commercial Nafion. Meanwhile, the optimal pore size endows graphdiyne membrane superior proton selectivity in aqueous solutions. Thus, as an experimentally fabricated material, graphdiyne membranes show a great potential as superior PEM materials in FCs, sensors, and other applications. The identification of superior proton conductivity and selectivity of graphdiyne membrane could be an important step in PEM studies and provide a avenue for the applications of nanomesh materials.

## Methods

**AIMD simulations**. Due to the delicate interaction of water, none of the existing functionals is able to universally and faultlessly describe water[41,42] and its self-ions[43–45] in various realistic conditions. In particular, the water–carbon interaction is demonstrated to be extremely difficult to simulate, even high-cost diffusion Monte Carlo, coupled cluster theory, and random phase approximation give rise to various adsorption energies of a single water molecule absorbed on graphene (Supplementary Tables 1 and 2 and Supplementary Note 5)[46–49]. Although dispersion correction remedies the bad performance of DFT in describing water–carbon adsorption interactions[46,47], relatively large variations are still found in interaction energies between a single water molecule and graphene predicted by different DFT models[49,50]. Despite these discrepancies, dispersion-corrected generalized gradient approximation (GGA) still gives rise to reasonable interfacial structures between liquid water and graphene[51,52] or carbon nanotubes[53], and describes well the monolayer ice on graphite[54].

Here, AIMD and PIMD simulations were performed using the i-PI program[55] for the dynamics and the CP2K code[56] for the calculation of first-principles energies and forces. We used the Becke–Lee–Yang–Parr (BLYP) exchange-correlation functional[57,58], a GGA functional, and the double-zeta valence polarized basis set and Goedecker–Teter–Hutter pseudopotentials[59,60]. The D3 empirical van der Waals corrections[61] were chosen to obtain a reasonable description of interactions between water and the membranes (Supplementary Fig. 12, Supplementary Note 6). Calculations with hybrid functional B3LYP-D3 verify that the choice of exchange-correlation functionals does not affect the qualitative conclusions of this work (Supplementary Note 4). We note that BLYP-D3 exhibits an overestimate of adsorption energy of a water molecule on graphene[49] and on graphdiyne here (Supplementary Fig. 12b); however, the characteristic interfacial water structure is well described in BLYP-D3 (Supplementary Figs. 8, 9, and 10), which is the cornerstone for proton transparency identified in the present work. All MD simulations were performed at 300 K with NVT ensembles and a timestep of 0.5 fs. Stochastic velocity rescaling thermostat[62] was used to control the temperature in AIMD simulations, while the PIGLET method[63] was used to

include NQEs in PIMD simulations. Six replicas per nuclei were used to exhibit the distribution of quantum nuclei in PIMD simulations.

Two sets of equilibrium AIMD simulations were performed with a hexagonal box with a dimension of $9.5 \times 9.5 \times 40$ Å to illuminate the diffusion behavior of protons at the interface. Total 32 water molecules were equally located on both sides of the graphdiyne membrane, with a $H_3O^+$ complex located in two positions initially corresponding to the two sets of simulations. The thicknesses of the water layers on the two sides are both about 8 Å. In the first set of simulations, the $H_3O^+$ complex is embedded in the bulk water layer below graphdiyne membrane, while in the second set of simulations, the $H_3O^+$ complex is located right beneath the nanopore of graphdiyne membrane. After optimizations, 4 and 6 AIMD simulations were performed for 15 and 10 ps to sample the diffusion process of proton at the interface.

AIMD and PIMD simulations with an electric field of 0.1 V Å$^{-1}$ were performed to simulate PT across graphdiyne membrane under non-equilibrium conditions and to illuminate the NQEs of proton transfer in a confined system. Within the same box, two $H_3O^+$ complexes were arrayed in a larger water layer of 28 water molecules below graphdiyne membrane, shown in Supplementary Fig. 5. After optimization, 10 AIMD and 10 PIMD simulations were performed to simulate PT across graphdiyne membrane independently. We note that the beads of nuclei were pre-equilibrated for 300 fs at 300 K before PIMD simulations.

**Energy barrier calculations**. CI-NEB[64,65] calculations including nine replicas were performed to calculate the energy barriers for proton transfer through 2D materials with the CP2K code. Two water molecules were equally located on both sides of membrane planes to include the aqueous effects in PT. The scanning-path methods were used to calculate the energy barriers for soluble fuel molecules and ions passing through graphdiyne membrane. Fuel molecules rigidly pass through the center of nanopore along length direction of molecules. Six extra water molecules were included to describe the hydration effects of ions. Membranes are fixed for all energy barrier calculations.

## Data availability
The data that support the findings of this study are available from the corresponding author upon reasonable request.

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

## Acknowledgements
We acknowledge financial support from Ministry of Science and Technology (Nos. 2016YFA0300902 and 2015CB921001), the National Natural Science Foundation of China (Nos. 11774396, 91850120 and 11934004), and Chinese Academy of Sciences (No. XDB070301).

## Author contributions
S.M. and E.G.W. conceived and directed the project. J.X. performed the AIMD and PIMD simulations and DFT calculations. J.X. and H.J. analyzed the data. Y.S., X.-Z.L., and E.G.W. participated in discussion. J.X., H.J., X.-Z.L., and S.M. wrote the paper with help from all co-authors.

## Additional information

**Competing interests:** The authors declare no competing interests.

