## [Peer Review File · Nature Communications]

Reviewers' comments:

Reviewer #1 (Remarks to the Author):

This study reports the results of ab initio molecular dynamics (AIMD) simulations of proton transport on graphdiyne. The computational protocol used in this study is standard and the simulations appear to be carried out in a careful way. The main contribution of this study is therefore the characterization and interpretation of the molecular mechanisms of proton transport in this system. As in any computer simulation, both these aspects depend critically on the model used in the simulations. In this context, all AIMD simulations reported in this study are carried out with the BLYP-D3 functional. There is mounting evidence that dispersion-corrected GGA functionals may not possess the necessary accuracy to correctly describe the properties of water in a faithful way, and meta-GGA and hybrid functionals are needed (e.g., see Chem. Rev., 2016, 116 (13), pp 7501–7528; J. Phys. Chem. Lett., 2018, 9, 5009; PNAS, 2019, 116, 1110).

In addition, to the best of my knowledge, the interaction of water with graphyne-like materials has not yet been characterized in any detail in the literature. Considering existing difficulties for electronic structure methods in describing the interactions of water with graphene (e.g., see how large are the variations in the DMC binding energies reported by the same group in two subsequent studies: Phys. Rev B 2011, 84, 033402 (2011) and J. Phys. Chem. Lett., 2019, 10, 358) and how different are the predictions from various DFT models (see <https://doi.org/10.26434/chemrxiv.7381943.v1>), it seems that benchmark calculations for water on graphyne-like systems would be necessary before moving to actual AIMD simulations.

In summary, this is a carefully done study. The main concern is that, to the best of my knowledge, there is no evidence that the DFT model used in the AIMD simulations has the necessary accuracy to provide a realistic representation of the graphdiyne-protonated water system that is studied here .

Reviewer #2 (Remarks to the Author):

The manuscript by J. Xu et al. addresses the transport of proton across the graphdiyne by means of ab initio calculations.

The topic is of great and wide interest for the many implications of this process in several applications from PEM fuel cells to semi-permeable membranes and so on. The manuscript focuses on the jump of a proton between two water molecules that are at the opposite side of one of the nanopores that are present in the graphdiyne structure. The investigation is carried out with standard ab initio molecular dynamics simulations coupled with static and dynamic characterization of the minimum-energy path for the proton transfer.

Among the merits of this study there is the careful study of different features concerning the water molecule dynamics and the nature of the solvated proton in two regions of the system under investigation (close and far from the graphdiyne); the selectivity for the proton against different ions; the role of quantum effects via path integral simulations.

Overall, the results show the graphdiyne is highly selective and transparent for proton transport, meaning that the proton jump has a very low activation energy, which is easily overcome at room temperature.

However, the work presents some flaws that merit further attention by the authors.

1. The level of theory employed (BLYP, a standard GGA density functional) is not suited to study proton transport, because it is well known that the self-interaction error in semi-local density

functional leads to fictitious delocalization of the electron density, which helps the proton jump and provide too low energy barrier for the H⁺ transfer reaction. See this work for a reference on the correct level of theory to be employed to refine the energetics of proton transfer reactions, *J. Chem. Theory Comput.*, 2012, 8, pp 3082–3088. I suggest the authors to run benchmark calculations with a hybrid HF-DFT density functional in order to set the error that is affecting the BLYP-based AIMD simulations.

2. The authors applied the semi-empirical D3 correction to DFT for taking into account the Dispersion interaction. This method has been developed and validated on isolated molecular complexes, is it enough accurate to describe water solution? is it enough accurate to describe water graphdiyne interaction? Test calculations or solid references on the application of BLYP-D3 to water graphdiyne or similar systems (water graphene) should be provided. See for example *J. Phys. Chem. Lett.*, 2019, 10, pp 358–368.

3. The proton transport in water has been always found to be very easy when the water molecules are in hydrophobic environment, as inside carbon nanotube [see for example *Nature Nanotech.* 2016, 11, pp 639–644; *Phys. Chem. Chem. Phys.* 2013, 15, pp 6344-6349]. The graphdiyne is not exception. However the author have not discussed extensively the correlation between the hydrophobic interaction, the dynamics of the water that are closest to the graphdiyne and the proton transport.

Responses to the reviewers' comments

First of all, we would like to express our thanks to the reviewers for the valuable comments, which are very helpful for us to improve the quality of the manuscript. The reviewer's comments are listed below, followed by our responses and changes made in the revised manuscript.

Reply to Referee 1

Comments: This study reports the results of ab initio molecular dynamics (AIMD) simulations of proton transport on graphdiyne. The computational protocol used in this study is standard and the simulations appear to be carried out in a careful way. The main contribution of this study is therefore the characterization and interpretation of the molecular mechanisms of proton transport in this system.

Response: We are grateful for the referee's recognition of our main findings and his/her assessment of our work being "carried out in a careful way". There are no critical technical problems about the manuscript. The referee has some concerns on the accuracy of DFT models to simulate graphdiyne-protonated water system. The comments and suggestions are all constructive and valuable.

We considered all the comments very carefully, and performed benchmark calculations with dispersion-corrected hybrid functionals. We demonstrate that the proton transparency property of graphdiyne membrane is independent of the choice of functionals.

Below we discuss these points one by one, along with corresponding changes we made in the revised text.

As in any computer simulation, both these aspects depend critically on the model used in the simulations. In this context, all AIMD simulations reported in this study are carried out with the BLYP-D3 functional. There is mounting evidence that dispersion-corrected GGA functionals may not possess the necessary accuracy to correctly describe the properties of water in a faithful way, and meta-GGA and hybrid

functionals are needed (e.g., see Chem. Rev., 2016, 116 (13), pp 7501–7528; J. Phys. Chem. Lett., 2018, 9, 5009; PNAS, 2019, 116, 1110).

Response: We thank the referee for the good reminder. Following his/her suggestions, we have performed benchmark calculations with both dispersion-corrected GGA functional and hybrid functional, and further justified the validity of major findings in this work.

We carefully read the papers Chem. Rev., 2016, 116 (13), pp 7501–7528; J. Phys. Chem. Lett., 2018, 9, 5009; PNAS, 2019, 116, 1110. The first paper reviews the development of analytical potential energy functions that represent many-body effects of water. The second paper highlights the performance of the meta-GGA functional B97M-rV, which display accuracy comparable to a dispersion-corrected hybrid functional, revPBE0-D3. The third shows the thermodynamic properties of liquid water and ices predicted by using hybrid functional, a machine-learning potential and advanced free-energy techniques.

To verify the independence of the findings in this work on the choice of functionals, we perform additional benchmark calculations using both dispersion-corrected GGA functional (BLYP-D3) and hybrid functional (B3LYP-D3). We calculated the energy barrier of proton transfer in free H_3O^+ - H_2O complex and transmembrane H_3O^+ - H_2O complex with both functionals (BLYP-D3 and B3LYP-D3). Figure S5a-b displays the models used. Figure S5c-d shows the energy barriers of two proton transfer processes calculated with the two functionals. The energy barrier of zero means the formation of stable H_5O_2^+ complex. Despite some numerical differences in the two functionals, both functionals give nearly the same energy profiles for proton transfer in free and transmembrane H_3O^+ - H_2O complex. **Both functionals (GGA and hybrid) demonstrated the negligible effect of graphdiyne membrane on proton transfer (namely, direct proton-graphdiyne interactions are not critical here), thus confirming the proton transparency property of graphdiyne membrane.** Despite that a little larger energy barrier is obtained in B3LYP-D3, we note that the proton transparency property is based on the comparison between the energy barrier in the two respective systems (with or without graphdiyne). Besides, the energy barrier of 0.27 eV calculated by the hybrid functional for transmembrane proton transfer is still much smaller than those in other 2D membranes.

Figure S5. Structural models of (a) free $\text{H}_3\text{O}^+\text{-H}_2\text{O}$ complex and (b) transmembrane $\text{H}_3\text{O}^+\text{-H}_2\text{O}$ complex. Energy barriers of proton transfer in free and transmembrane $\text{H}_3\text{O}^+\text{-H}_2\text{O}$ complex calculated with (c) BLYP-D3 and (d) B3LYP-D3.

To further check the effect of aqueous environment in benchmark calculations, we extract from the trajectories in *ab initio* molecular dynamics simulations a snapshot in which proton diffuses close to graphdiyne nanopore, and perform geometry optimization with hybrid functional B3LYP-D3. Figure S8 shows the optimized structure and charge density distribution of the system. The OO distance is 2.52 \AA and the angle H-O...O is 2°. The transmembrane hydrogen bond is explicitly formed across graphdiyne membrane. Thus the existence of proton transport channel is confirmed with hybrid B3LYP-D3, which is the cornerstone of proton transparency property. The excess proton in the transmembrane hydrogen bond has d_{OO} of 2.52 \AA and δ of 0.39 \AA , in good agreement with the features of H_3O^+ complex shown in Figure 4b calculated by BLYP-D3 functional ($d_{\text{OO}}=2.52$ \AA and $\delta=0.36$ \AA).

Figure S8. Optimized structure with B3LYP-D3 and the charge density distribution of graphdiyne immersed in water.

Despite improved accuracy of the hybrid functionals, the heavy computational costs are prohibitive for their efficient use in long-time ab initio molecular dynamics simulations. Indeed, the GGA functionals are widely used in previous studies of proton transport in liquid water. For example, BLYP is used in simulating proton transfer through water gossamer (PNAS 2013, 110, 13723–13728), hydrogen bond fluctuations in water (PNAS 2013, 110, 15591–15596), nuclear quantum effects in H⁺ and OH⁻ diffusion along confined water wires (J. Phys. Chem. Lett. 2016, 7, 3001–3007), and proton transport in biological systems (PNAS 2017, 114, 13182). GGA is also used in studies of nanoconfinement effects on hydrated excess protons (Nat. Commun. 2013, 4, 2349) and proton holes (Nat. Commun. 2016, 7, 12625), proton transfer across single-layer graphene (Nat. Commun. 2015, 6, 6539; J. Phys. Chem. Lett. 2017, 8, 4354), proton transfer on substrates, e.g., metal (PRL 2017, 119, 126001) and hydroxylated graphene (PRL 2017, 118, 186101; J. Phys. Chem. Lett. 2019, 10, 518).

We have made the following changes in the text.

i) Add the following sentence at the top of page 8:

“Despite the small difference, the same energy profiles demonstrate that the TM PT is easily driven by thermal fluctuations, in the same manner to that in bulk water.”

ii) Add the following sentences in the middle of page 11:

“The TM HBs across graphdiyne offer a fast channel for proton transfer in the Grotthuss mechanism, which accounts for the proton transparency property of graphdiyne membrane. The proton transparency property is further demonstrated with nearly the same energy barrier profiles of proton transfer in free and transmembrane H₃O⁺-H₂O complexes (Figure S5c). Besides, benchmark calculations with the hybrid functional confirm the proton transparency property of graphdiyne membrane (Figure S5d, Figure S6, Figure S7) and the existence of TM HBs (Figure S8).”

iii) Add the following sentence in the middle of page 13:

“Benchmark calculations with hybrid functional B3LYP verify that the choice of exchange-correlation functionals does not affect the qualitative conclusions of this work (see supporting information).”

iv) Add Section 5 with 7 paragraphs and 4 figures, Figure S5, Figure S6, Figure S7, and Figure S8, in supporting information to introduce benchmark calculations with the hybrid functional and to clarify the independence of proton transparency from the choice of functionals.

In addition, to the best of my knowledge, the interaction of water with graphyne-like materials has not yet been characterized in any detail in the literature. Considering existing difficulties for electronic structure methods in describing the interactions of water with graphene (e.g., see how large are the variations in the DMC binding energies reported by the same group in two subsequent studies: Phys. Rev B 2011, 84, 033402 (2011) and J. Phys. Chem. Lett., 2019, 10, 358) and how different are the predictions from various DFT models (see <https://doi.org/10.26434/chemrxiv.7381943.v1>), it seems that benchmark calculations for water on graphyne-like systems would be necessary before moving to actual AIMD simulations.

Response: Thank the referee for the suggestion. Following his/her suggestion, we have performed additional benchmark calculations on water-graphyne interactions.

We have carefully checked the papers Phys. Rev B 2011, 84, 033402, J. Phys. Chem. Lett., 2019, 10, 358, <https://doi.org/10.26434/chemrxiv.7381943.v1>. The first paper presents the adsorption of water monomer on graphene with various methods from LDA to RPA and DMC. The second paper from the same group reports extensive benchmark calculations of water monomer adsorbed on benzene, coronene and graphene with DMC, CCSD(T) and RPA methods. The third paper shows the water monomer adsorbed on polycyclic aromatic hydrocarbons with various methods from GGA to RPA, DMC and CCSD(T). All three papers give systematic calculations of water adsorption on graphene.

We summarize the results on water-graphene and water-graphyne interactions as calculated from different methods in Fig. S10. Figure S10a exhibits the energy profiles of water on graphene, which are extracted from literature (PRB 2011, 84,

033402; J. Phys. Chem. Lett., 2019, 10, 358). We see that BLYP-D exhibits an energy profile consistent with those of accurate DMC calculations. In this regard, dispersion-correction is crucial for the simulation of water adsorption behavior. Consequently, dispersion-corrected DFT were recommended for efficient description of water-graphene interaction. Although large variations in the interaction energies predicted by various DFT models were presented in <https://doi.org/10.26434/chemrxiv.7381943.v1>, BLYP-D3 still gives the results in the correct range of high-level electronic structure methods.

Figure S10. Energy profiles of water adsorption on (a) graphene and (b) graphdiyne with one OH bond pointing to the surface. The inset exhibits the adsorption structures. The BLYP, B3LYP and BLYP-D results of water adsorption on graphene are taken from Phys. Rev B 2011, 84, 033402, and DMC results from J. Phys. Chem. Lett., 2019, 10, 358-368, while all other data are calculated in the present work.

Following the referee's suggestion, we also performed benchmark calculations on the water-graphdiyne interaction. Figure S10b shows the energy profiles of the water molecule approaching the center of nanopore in the graphdiyne membrane. Bare BLYP and B3LYP both give rise to a repulsive interaction. Once dispersion correction (D3) is included, the adsorption behavior is recovered for both BLYP-D3 and B3LYP-D3. Despite some numerical differences, the two functionals give qualitatively similar results. Moreover, the energy profile and atomic structure in optimized water dimer complexes with and without graphdiyne are nearly the same (Figure S6 and S7). Thus, BLYP-D3 gives a proper description of interfacial water structure, attaining the accuracy of B3LYP-D3. Based on these studies, dispersion-

corrected BLYP is a reasonable choice to study the interaction between water molecules and carbon materials.

To be more specific, we note that consistent energetics were obtained from both functionals to describe free and transmembrane proton complexes with a varying OO distance d_{OO} (Figure S6 and Figure S7). The presence of graphdiyne membrane show a negligible effect for both functionals, suggesting the water-graphdiyne interactions have been adequately addressed.

Figure S6. Energy profile as the function of OO distance in (a) free and (b) transmembrane proton complex.

Figure S7. Optimized transmembrane proton complexes calculated by (a) BLYP-D3 and (b) B3LYP-D3 functional. Optimized transmembrane $(H_2O)_2$ structures calculated by (c) BLYP-D3 and (d) B3LYP-D3 functional. Both functionals give similar water structures.

We have made the following changes in the text:

- i) Add the following sentences in the middle of page 11:

“The TM HBs across graphdiyne offer a fast channel for proton transfer in the Grotthuss mechanism, which accounts for the proton transparency property of graphdiyne membrane. The proton transparency property is further demonstrated with nearly the same energy barrier profiles of proton transfer in free and transmembrane H_3O^+ - H_2O complexes (Figure S5c). Besides, benchmark calculations with the hybrid functional confirm the proton transparency property of graphdiyne membrane (Figure S5d, Figure S6, Figure S7) and the existence of TM HBs (Figure S8).”

ii) Add the following sentence in the middle of page 13:

“Benchmark calculations with hybrid functional B3LYP verify that the choice of exchange-correlation functionals does not affect the qualitative conclusions of this work (see supporting information).”

iii) Add Section 5 with 7 paragraphs and 4 figures, Figure S5, Figure S6, Figure S7, and Figure S8, in supporting information to introduce the benchmark calculations with hybrid functional and to clarify the independence of proton transparency from the choice of functionals.

In summary, this is a carefully done study. The main concern is that, to the best of my knowledge, there is no evidence that the DFT model used in the AIMD simulations has the necessary accuracy to provide a realistic representation of the graphdiyne-protonated water system that is studied here.

Response: We thank the referee for his/her the assessment “a carefully done study”. To verify the validity of DFT models, we performed benchmark calculations with dispersion-corrected hybrid functional B3LYP-D3. The negligible effects of graphdiyne on transmembrane proton transfer is confirmed, further justifying the proton transparency property of graphdiyne—the major conclusion of the present work (Figures S5-S8, S10). Besides, despite being less accurate, the GGA functionals (BLYP and PBE) are widely used in previous studies of proton transport in protonated water. In particular, dispersion-corrected functionals are demonstrated to give a reasonable description of interactions between water molecules and carbon materials, close to that of high-level calculations such as DMC.

Reply to Referee 2

Comments: The manuscript by J. Xu et al. addresses the transport of proton across the graphdiyne by means of ab initio calculations.

The topic is of great and wide interest for the many implications of this process in several applications from PEM fuel cells to semi-permeable membranes and so on. The manuscript focuses on the jump of a proton between two water molecules that are at the opposite side of one of the nanopores that are present in the graphdiyne structure. The investigation is carried out with standard ab initio molecular dynamics simulations coupled with static and dynamic characterization of the minimum-energy path for the proton transfer.

Among the merits of this study there is the careful study of different features concerning the water molecule dynamics and the nature of the solvated proton in two regions of the system under investigation (close and far from the graphdiyne); the selectivity for the proton against different ions; the role of quantum effects via path integral simulations.

Overall, the results show the graphdiyne is highly selective and transparent for proton transport, meaning that the proton jump has a very low activation energy, which is easily overcome at room temperature.

Response: We greatly appreciate the referee's recognition of our key findings and their practical significance. The referee also makes constructive suggestions to further improve the present work, though he/she has some doubts about the DFT model. Comments and suggestions are all constructive and valuable.

We considered all the comments very carefully, and performed benchmark calculations with dispersion-corrected hybrid functional. We demonstrate that the proton transparency property of graphdiyne membrane is independent of the choice of functional. We give the evidences of the necessity of dispersion-correction in describing water and carbon materials. The hydrophobic effect of carbon materials is discussed, and the comparison between graphdiyne nanopore and carbon nanotube is given.

Below we discuss the detailed questions one by one, along with corresponding changes we made in the revised text.

However, the work presents some flaws that merit further attention by the authors.

1. The level of theory employed (BLYP, a standard GGA density functional) is not suited to study proton transport, because it is well known that the self-interaction error in semi-local density functional leads to fictitious delocalization of the electron density, which helps the proton jump and provide too low energy barrier for the H⁺ transfer reaction. See this work for a reference on the correct level of theory to be employed to refine the energetics of proton transfer reactions, J. Chem. Theory Comput., 2012, 8, pp 3082–3088. I suggest the authors to run benchmark calculations with a hybrid HF-DFT density functional in order to set the error that is affecting the BLYP-based AIMD simulations.

Response: Thank referee for the nice suggestion. We carefully read the paper J. Chem. Theory Comput., 2012, 8, pp 3082–3088. The paper presents a systematic study on proton transfer reaction in various systems with functionals ranging from GGA to hybrids, where hybrid functionals show an improved accuracy.

Following the referee's suggestion, we performed benchmark calculations with dispersion-corrected hybrid functional B3LYP-D3.

(1) We calculated the energy barrier of proton transfer in free and transmembrane H₃O⁺-H₂O complex with both functionals (BLYP-D3 and B3LYP-D3). Figure S5a-b displays the models used. Figure S5c-d shows the energy barriers of two proton transfer processes calculated with the two functionals. The energy barrier of zero means the formation of stable H₅O₂⁺ complex. Despite some numerical differences in the two functionals, both functionals give nearly the same energy profiles for proton transfer in free and transmembrane H₃O⁺-H₂O complex. **Both functionals (GGA and hybrid) demonstrated the negligible effect of graphdiyne membrane on proton transfer (namely, direct proton-graphdiyne interactions are not critical here), thus confirming the proton transparency property of graphdiyne membrane.** Despite that a little larger energy barrier is obtained in B3LYP-D3, we note that the proton transparency property is based on the comparison between the energy barrier in the two respective systems (with or without graphdiyne). Besides, the energy

barrier of 0.27 eV calculated by the hybrid functional for transmembrane proton transfer is still much smaller than those in other 2D membranes.

Figure S5. Structural models of (a) free $\text{H}_3\text{O}^+\text{-H}_2\text{O}$ complex and (b) transmembrane $\text{H}_3\text{O}^+\text{-H}_2\text{O}$ complex. Energy barriers of proton transfer in free and transmembrane $\text{H}_3\text{O}^+\text{-H}_2\text{O}$ complex calculated with (c) BLYP-D3 and (d) B3LYP-D3.

(2) To further check the effect of aqueous environment in benchmark calculations, we extract from the trajectories in *ab initio* molecular dynamics simulations a snapshot in which proton diffuses close to graphdiyne nanopore, and perform geometry optimization with hybrid functional B3LYP-D3. Figure S8 shows the optimized structure and charge density distribution of the system. The OO distance is 2.52 \AA and the angle H-O...O is 2°. The transmembrane hydrogen bond is explicitly formed across graphdiyne membrane. Thus the existence of proton transport channel is confirmed with hybrid B3LYP-D3, which is the cornerstone of proton transparency property. The excess proton in the transmembrane hydrogen bond has d_{OO} of 2.52 \AA and δ of 0.39 \AA , in good agreement with the features of H_3O^+ complex shown in Figure 4b calculated by BLYP-D3 functional ($d_{\text{OO}}=2.52$ \AA and $\delta=0.36$ \AA).

(3) We also performed benchmark calculations on the water-graphdiyne interaction. Figure S10b shows the energy profiles of the water molecule approaching the center of nanopore in the graphdiyne membrane. Despite some numerical differences, both BLYP-D3 and B3LYP-D3 give qualitatively similar results. Moreover, the energy

profile and atomic structure in optimized water dimer complexes with and without graphdiyne are nearly the same (Figure S6 and S7). Thus, BLYP-D3 gives a proper description of interfacial water structure, attaining the accuracy of B3LYP-D3.

Figure S10. Energy profiles of water adsorption on (a) graphene and (b) graphdiyne with one OH bond pointing to the surface. The inset exhibits the adsorption structures. The BLYP, B3LYP and BLYP-D results of water adsorption on graphene are taken from Phys. Rev B 2011, 84, 033402, and DMC results from J. Phys. Chem. Lett., 2019, 10, 358-368, while all other data are calculated in the present work.

Figure S6. Energy profile as the function of OO distance in (a) free and (b) transmembrane proton complex.

We have made the following changes in the text.

i) Add the following sentence at the top of page 8:

“Despite the small difference, the same energy profiles demonstrate that the TM PT is easily driven by thermal fluctuations, in the same manner to that in bulk water.”

ii) Add the following sentences in the middle of page 11:

“The TM HBs across graphdiyne offer a fast channel for proton transfer in the Grotthuss mechanism, which accounts for the proton transparency property of graphdiyne membrane. The proton transparency property is further demonstrated with nearly the same energy barrier profiles of proton transfer in free and transmembrane H_3O^+ - H_2O complexes (Figure S5c). Besides, benchmark calculations with the hybrid functional confirm the proton transparency property of graphdiyne membrane (Figure S5d, Figure S6, Figure S7) and the existence of TM HBs (Figure S8).”

iii) Add the following sentence in the middle of page 13:

“Benchmark calculations with hybrid functional B3LYP verify that the choice of exchange-correlation functionals does not affect the qualitative conclusions of this work (see supporting information).”

iv) Add Section 5 with 7 paragraphs and 4 figures, Figure S5, Figure S6, Figure S7, and Figure S8, in supporting information to introduce the benchmark calculations with the hybrid functional and to clarify the independence of proton transparency from the choice of functionals.

2. The authors applied the semi-empirical D3 correction to DFT for taking into account the Dispersion interaction. This method has been developed and validated on isolated molecular complexes, is it enough accurate to describe water solution? is it enough accurate to describe water graphdiyne interaction? Test calculations or solid references on the application of BLYP-D3 to water graphdiyne or similar systems (water graphene) should be provided. See for example J. Phys. Chem. Lett., 2019, 10, pp 358–368.

Response: Thank the referee for the comment. We carefully read the papers J. Phys. Chem. Lett., 2019, 10, 358-368. The paper presents extensive benchmark calculations of water adsorption on benzene, coronene and graphene with DMC, CCSD(T) and RPA methods.

Despite being developed on isolated molecular complexes, D3 vdW correction is demonstrated to be crucial for the intrinsic properties of water, e.g., water density maximum and negative volume of melting (PNAS 2016, 113, 8368). Besides, the BLYP-D3 simulations give rise to the liquid water structure closer to experimental data than bare BLYP (J. Chem. Theory Comput. 2012, 8, 3902–3910). Among dispersion-corrected GGA and vdW density functionals, the combination of BLYP with D3 provide the best agreement with experiment on the dipole moment and diffusion coefficient of liquid water (J. Phys. Chem. C 2014, 118, 29401–29411). Furthermore, BLYP-D3 is used in simulating water solutions and solutes, e.g., criegee (J. Am. Chem. Soc. 2018, 140, 4913).

In the meantime, dispersion-correction is demonstrated to remedy the bad performance of DFT in describing water-carbon interactions. Figure S10a exhibits the energy profile of water adsorption on graphene calculated by BLYP-D, consistent well with that by DMC (Phys. Rev B 2011, 84, 033402; J. Phys. Chem. Lett., 2019, 10, 358-368). Without dispersion-correction, repulsive interaction is obtained in both GGA and hybrid functionals (Phys. Rev B 2011, 84, 033402). We also performed benchmark calculations of water adsorbed on the nanopore of graphdiyne membrane with and without D3. As shown in Figure S10b, without D3 correction, repulsive interaction is obtained with both BLYP and B3LYP. Once D3 is included, the adsorption behavior is recovered with both functionals.

Therefore, dispersion-corrected DFT gives reasonable description of water-water and water-carbon interactions, which necessitate its usage in the present work.

We have made the following changes in the text:

i) Revise the following sentences in the middle of page 13:

“The D3 empirical Van der Waals corrections⁴⁷ were chosen to obtain a reasonable description of interactions between water and the membranes.⁴⁸⁻⁴⁹ Dispersion-corrected DFT is demonstrated to exhibit nice performance in describing the water-carbon adsorption systems (Figure S10).”

ii) Add Section 7 in supporting information to illustrate the importance of introduction of dispersion-correction in DFT calculations.

3. The proton transport in water has been always found to be very easy when the water molecules are in hydrophobic environment, as inside carbon nanotube [see for example Nature Nanotech. 2016, 11, pp 639–644; Phys. Chem. Chem. Phys. 2013, 15, pp 6344-6349]. The graphdiyne is not exception. However the author have not discussed extensively the correlation between the hydrophobic interaction, the dynamics of the water that are closest to the graphdiyne and the proton transport.

Response: Thank referee for the suggestion. In the revised manuscript, we carefully discuss the correlation between the hydrophobic water-graphdiyne interaction and the dynamics of proton transfer.

We have carefully read the papers Nature Nanotech. 2016, 11, pp 639–644; Phys. Chem. Chem. Phys. 2013, 15, pp 6344-6349. The first paper shows the superior proton transport rate exceeding to that of bulk water in 0.8-nm-diameter carbon nanotube porins, which is attributed to the formation of one-dimensional water wires. The second paper, written by some of us (X.Z.L. and E.G.W.), presents the enhancement of proton transport rate by the water wire in carbon nanotube, and highlights the enhancement of proton transport rate by quantum nuclear effects. The hydrophobic interaction between water molecule and narrow carbon nanotube, e.g., 0.8-nm-diameter carbon nanotubes, leads to weak interaction between them and facilitates the formation of one-dimensional water wire inside the nanotube. Proton transport along the single water wire is in Grotthuss mechanism, thus superior proton transport rate is obtained.

The adequate size of 0.55 nm of nanopore on graphdiyne membrane allows the formation of transmembrane hydrogen bonds between the water molecules on both sides. The excess proton does not bond to the pore rim (Figure S8). The inert and neutral pore rim only serves as a spatial constraint, thus facilitating the formation of transmembrane hydrogen bonds. Here, the inert and neutral pore rim nicely reflects its hydrophobic nature, as that for the hydrophobic inner wall of carbon nanotubes. Furthermore, the unique transmembrane structure can be regarded as the shortest water wire composed of only two water molecules. The Grotthuss mechanism of proton transport is demonstrated in our *ab initio* molecular dynamics simulations.

We have made the following changes in the text:

i) Revise the following sentence at the top of page 7:

“We note that the adequate pore size and the hydrophobic pore rim is the key to the formation of stable TM HBs.”

ii) Revise the following sentence at the top of page 8:

“Although proton diffusion involves breaking of OH bonds, the excess proton could not bond to graphdiyne membrane (see Figure S1), which is attributed to the hydrophobic effects of the inert and neutral pore rim.”

iii) Add the following sentences in the middle of page 11:

“We note that the transmembrane structures can be regarded as the shortest one-dimensional water wires as those in 0.8-nm-diameter carbon nanotubes, where superior proton transport rate is also obtained.⁴⁰ The hydrophobic pore rim facilitates the formation of transport channel of this short one-dimensional water wire in graphdiyne nanopore, in the same way as that in the hydrophobic inner wall of carbon nanotubes.”

Reviewers' comments:

Reviewer #1 (Remarks to the Author):

The authors have repeated some of the calculations with B3LYP-D3. A comparison between the energy profiles obtained at the BLYP-D3 and B3LYP-D3 levels of theory is shown in Figure S10b. I think there are at least two qualitative differences: 1) the shape of the two profiles is different, 2) the position of the minimum is noticeably different, with B3LYP-D3 predicting a larger separation (by ~ 0.5 Å). These differences should be discussed since they may play an important role in the mechanism of proton transport across the membrane.

I still believe that higher-level calculations are required for a quantitative understanding of this process. Based on the DFT analysis shown in Fig. 9 of *J. Chem. Theory Comput.* 2019, 15, 2359, it seems that both BLYP-D3 and B3LYP-D3 (significantly) overestimate the interaction energies for water on graphene. Although a similar analysis for water on graphdiyne has not been reported yet, the similarity between the two systems may suggest similar trends, which may change the overall molecular picture of proton transport through graphdiyne.

Based on my knowledge and understanding, I think we are still far from an accurate determination of how water interacts with graphitic-like materials. A few years ago, Michaelides and coworkers reported DMC results for water on graphene (*Phys. Rev. B* 2011, 84, 033402) that have been considered as the "gold standard" until the same group reported a few months ago (*J. Phys. Chem. Lett.* 2019, 10, 358) a new set of DMC results that are significantly different from their original results. At the same time, another group has carried out an extensive and systematic analysis using different functionals (*J. Chem. Theory Comput.* 2019, 15, 2359) which shows that different functionals predict significantly different binding energies between water and graphene.

To this, one should add that it is becoming evident that none of the existing functionals is able to correctly describe not only water (e.g., see Section 4.3, Figs. 8, 9, 10, and Table 4 of our *Chem. Rev.* 2016, 116, 7501, as well a recent analysis of many-body effects in water using DFT: <https://doi.org/10.26434/chemrxiv.8026553.v1>) but also its self-ions (e.g., recent studies: *J. Chem. Theory Comput.* 2018, 14, 1982 and <https://doi.org/10.26434/chemrxiv.8068793.v1>).

I am probably biased but I think it is hard to make a case for the reliability of the calculations presented in this study based on what is known. This doesn't mean that the results and the mechanisms derived from the simulations are "wrong" because error cancellation may compensate known deficiencies of the functionals and lead, fortuitously, to the "correct" representation of the process under investigation. In a sense, the simulations may be "right" for the wrong reasons.

In terms of calculations that the authors may want to consider to carry out, I would say that some DMC and L-CCSD(T) calculations may be helpful. However, these calculations are very expensive and will probably take months, which may not be appealing.

In summary, this manuscript reports interesting and intriguing results. Unfortunately, based on the available data, I think it is not possible to unambiguously validate the theoretical results, which may turn out to be incredible predictions as well as just artifacts due to the limitations that affect current DFT models.

Reviewer #2 (Remarks to the Author):

I am fine with the replies to my concerns, the authors performed a benchmark study that provides

solid foundations for the quality of reported DFT results. For these reasons I suggest publication of the revised manuscript.

Responses to the reviewers' comments

First of all, we would like to express our thanks to the reviewers for the valuable comments, which are very helpful for us to improve the quality of the manuscript. The reviewer's comments are listed below, followed by our responses and changes made in the revised manuscript.

Reply to Reviewer 1

Comments: The authors have repeated some of the calculations with B3LYP-D3. A comparison between the energy profiles obtained at the BLYP-D3 and B3LYP-D3 levels of theory is shown in Figure S10b. I think there are at least two qualitative differences: 1) the shape of the two profiles is different, 2) the position of the minimum is noticeably different, with B3LYP-D3 predicting a larger separation (by ~0.5 Å). These differences should be discussed since they may play an important role in the mechanism of proton transport across the membrane.

Response: Thank the reviewer for the good comments. Indeed, the energy profiles of **1-leg configurations** of single water molecule exhibit some different shapes and minimum positions for different functionals in Figure S10b; however, this difference diminishes when more water molecules are included thanks to the many-body effect.

First, we note that, since the transmembrane hydrogen bonds are formed with 0-leg and 1-leg water molecules on both sides of graphdiyne membrane, the energy profiles of 0-leg configurations of water molecule are also calculated. Nearly the same minimum positions are obtained for the 0-leg configurations, as shown in Figure S10b.

In the meantime, we considered the many-body effects in the energy profiles by introducing another water molecule on the other side to form the transmembrane hydrogen bond. The optimized transmembrane (H₂O)₂ complexes exhibit nearly the same structures from both BLYP-D3 and B3LYP-D3 functionals, as shown in Figure S7c-d. The nearly same structures are also demonstrated in optimized transmembrane H₅O₂⁺ complexes in Figure S7a-b. We performed extra calculations with the two water molecule structures in Figure S7d. We fixed the left water molecule in Figure S7d and calculated the energy profiles of the right water molecule with both functionals, then

calculated the energy profiles of the left water molecule in the same way. As shown in Figure S7e-f, the energy profiles of the right and left water molecule exhibit the same shapes and minimum positions of $\sim 1.5 \text{ \AA}$ with both functionals. The small difference in the shape of energy profile diminishes due to the presence of many-body effects. We note that BLYP-D3 overestimates the interaction energy by $\sim 0.14 \text{ eV}$. However, the minimum position, which reflects the characteristic interfacial water structure and emergence of transmembrane hydrogen bonds, remains almost the same as that from the B3LYP-D3 calculations. It is the cornerstone of proton transparency property in present work.

Despite being less accurate, dispersion-corrected GGA indeed gives rise to reasonable interfacial structures between water and graphene, and between water and carbon-nanotubes in aqueous solutions (J. Phys. Chem. Lett. 2019, 10, 329–334; ACS Nano 2012, 6, 2401; PNAS 2015, 112, 10851–10856). Besides, the proton complex is also well simulated at liquid-graphdiyne interfaces here (Figure S8). Given the performance remediation of dispersion-corrected GGA in many-body systems, we think that the discrepancies of different functionals may decrease at the liquid-solid interfaces, and reasonable interfacial water structures can be obtained with dispersion-corrected GGA. We conclude that many-body effects play an important role in water-carbon adsorption systems.

Figure S10b. Energy profiles of water adsorption on graphdiyne. The inset exhibits the adsorption structures, and the water molecules above and below graphdiyne membrane are defined as 1-leg and 0-leg molecules, respectively.

Figure S7. Optimized transmembrane proton complexes calculated by (a) BLYP-D3 and (b) B3LYP-D3 functional. Optimized transmembrane (H₂O)₂ structures calculated by (c) BLYP-D3 and (d) B3LYP-D3 functional. Both functionals give similar water structures. Energy profiles of the right (e) and left (f) water molecule in transmembrane (H₂O)₂ structure in Figure S7d.

We have made the following changes in the text:

- i) Append energy profiles of 0-leg configuration of water molecule in Figure S10b.
- ii) Append Figure S7e-f in the supporting information.
- iii) Add the following sentences at the bottom of page 6 in supporting information.

“We performed additional calculations of the two water molecule structures in Figure S7d. We fixed the left water molecule in Figure S7d and calculated the energy profiles of the right water molecule with both functionals, then calculated the energy profiles of the left water molecule in the same way. As shown in Figure S7e-f, nearly the same minimum positions are obtained with both functionals for the left and right molecule respectively. Despite BLYP-D3 overestimates the interaction energy by ~0.14 eV, we note that the minimum position, which corresponds to the OO distance and reflects the characteristic interfacial water structure dominating transmembrane proton transport, remains the same for both functionals.”

iv) Add the following sentences at the top of page 11 in supporting information.

“Although the minimum positions are different by ~ 0.5 Å in 1-leg configurations for two dispersion-corrected functionals, the minimum positions are nearly the same in 0-leg configurations. Besides, the minimum positions are also nearly the same for adsorption of water molecules on both sides of the membrane (Figure S7e-f), which reflects the same interfacial water structure. Furthermore, dispersion-corrected GGA indeed gives rise to reasonable interfacial structures between water and graphene or carbon-nanotubes in aqueous solutions.⁸⁻¹⁰ Given the performance remediation of BLYP-D3 in many-body systems, we think that the discrepancies of different functionals may decrease due to the aqueous effects at liquid-solid interfaces, at least in the transmembrane adsorption system here. We infer that many-body effects play an important role in water-carbon interactions.”

v) Add the following sentences at the top of page 14 in the main text.

“We note that BLYP-D3 exhibits an overestimate of adsorption energy of a water molecule on graphene⁴⁹ and on graphdiyne here (Figure S10b), however the characteristic interfacial water structure is well described in BLYP-D3 (Figure S6, Figure S7, Figure S8), which is the cornerstone for proton transparency identified in the present work.”

vi) Add the following sentences in the middle of page 13 in the main text.

“Despite of these discrepancies, dispersion-corrected GGA still gives rise to reasonable interfacial structures between liquid water and graphene⁵¹⁻⁵² or carbon-nanotubes,⁵³ and describes well the monolayer ice on graphite.⁵⁴”

I still believe that higher-level calculations are required for a quantitative understanding of this process. Based on the DFT analysis shown in Fig. 9 of J. Chem. Theory Comput. 2019, 15, 2359, it seems that both BLYP-D3 and B3LYP-D3 (significantly) overestimate the interaction energies for water on graphene. Although a similar analysis for water on graphdiyne has not been reported yet, the similarity between the two systems may suggest similar trends, which may change the overall molecular picture of proton transport through graphdiyne.

Response: Thank the reviewer for the comment. We carefully read the paper J. Chem. Theory Comput. 2019, 15, 2359. In the paper, BLYP-D3 shows nice performance for 0-leg configuration of water molecule adsorption on graphene, while overestimates the adsorption energies for 1-leg and 2-leg configurations of water molecule on graphene. In fact, the overestimate of interactions for 1-leg and 2-leg is found for most of the DFT models. We summarized the adsorption behaviors of water molecule on graphene with different models in the supporting information.

We performed new calculations with PBE0-D3. As shown in Figure R1, the energy profile of PBE0-D3 is nearly the same as that of B3LYP-D3, showing that the B3LYP-D3 functional shows high consistency and can be regarded as the benchmark in the present work.

As per above discussion, BLYP-D3 indeed overestimates the adsorption energy of water molecule (Figure S7 and Figure S10), while the characteristic interfacial water structure is nicely described in BLYP-D3 (Figure S6, Figure S7, Figure S8), due to the presence of many-body effects.

Figure R1. Energy profiles of 1-leg and 0-leg configurations of a single water molecule adsorption on the nanopore of graphdiyne membrane.

We have made the following changes in the text:

i) Add the following sentences at the top of page 14 in the main text.

“We note that BLYP-D3 exhibits an overestimate of adsorption energy of a water molecule on graphene⁴⁹ and on graphdiyne here (Figure S10b), however the characteristic interfacial water structure is well described in BLYP-D3 (Figure S6, Figure S7, Figure S8), which is the cornerstone for proton transparency identified in the present work.”

ii) Add the following sentences in the middle of page 13 in the main text.

“Despite of these discrepancies, dispersion-corrected GGA still gives rise to reasonable interfacial structures between liquid water and graphene⁵¹⁻⁵² or carbon-nanotubes,⁵³ and describes well the monolayer ice on graphite.⁵⁴”

iii) Add the following sentences at the bottom of page 6 in supporting information.

“We performed additional calculations of the two water molecule structures in Figure S7d. We fixed the left water molecule in Figure S7d and calculated the energy profiles of the right water molecule with both functionals, then calculated the energy profiles of the left water molecule in the same way. As shown in Figure S7e-f, nearly the same minimum positions are obtained with both functionals for the left and right molecule respectively. Despite BLYP-D3 overestimates the interaction energy by ~0.14 eV, we note that the minimum position, which corresponds to the OO distance and reflects the characteristic interfacial water structure dominating transmembrane proton transport, remains the same for both functionals.”

iv) Insert Section 7 and Table S1 and S2 in supporting information to summarize the adsorption behaviors of water molecule on graphene, particularly highlighting the comparisons to high-level DMC and CCSD(T) results.

Based on my knowledge and understanding, I think we are still far from an accurate determination of how water interacts with graphitic-like materials. A few years ago, Michaelides and coworkers reported DMC results for water on graphene (Phys. Rev. B 2011, 84, 033402) that have been considered as the "gold standard" until the same group reported a few months ago (J. Phys. Chem. Lett. 2019, 10, 358) a new set of DMC results that are significantly different from their original results. At the same time, another group has carried out an extensive and systematic analysis using different functionals (J. Chem. Theory Comput. 2019, 15, 2359) which shows that different functionals predict significantly different binding energies between water and graphene.

To this, one should add that it is becoming evident that none of the existing functionals is able to correctly describe not only water (e.g., see Section 4.3, Figs. 8, 9, 10, and

Table 4 of our Chem. Rev. 2016, 116, 7501, as well a recent analysis of many-body effects in water using DFT: <https://doi.org/10.26434/chemrxiv.8026553.v1>) but also its self-ions (e.g., recent studies: J. Chem. Theory Comput. 2018, 14, 1982 and <https://doi.org/10.26434/chemrxiv.8068793.v1>).

I am probably biased but I think it is hard to make a case for the reliability of the calculations presented in this study based on what is known. This doesn't mean that the results and the mechanisms derived from the simulations are "wrong" because error cancellation may compensate known deficiencies of the functionals and lead, fortuitously, to the "correct" representation of the process under investigation. In a sense, the simulations may be "right" for the wrong reasons.

In terms of calculations that the authors may want to consider to carry out, I would say that some DMC and L-CCSD(T) calculations may be helpful. However, these calculations are very expensive and will probably take months, which may not be appealing.

In summary, this manuscript reports interesting and intriguing results. Unfortunately, based on the available data, I think it is not possible to unambiguously validate the theoretical results, which may turn out to be incredible predictions as well as just artifacts due to the limitations that affect current DFT models.

Response: Thank reviewer for the insightful comment. We carefully read these papers above. In the revised manuscript, we present in detail the complexities and controversies of simulations of water and its self-ions. The adsorption behaviors of water on graphene with high-cost many-body methods are also summarized in the supporting information.

Table S1 and S2 display the adsorption distances and adsorption energies of water molecules on graphene with 0-leg and 1-leg configurations respectively. The high-cost diffusion Monte Carlo, coupled cluster theory and random phase approximation give rise to various adsorption energies of a single water molecule absorbed on graphene for the two configurations. For the 0-leg configuration, BLYP-D3 shows the nice performances in describing this configuration. For the 1-leg configuration, overestimate of the adsorption energies is found for most DFT models, which is demonstrated in Ref.

4 and Ref. 7. Despite of the discrepancies of adsorption energies, the adsorption distances converge well for the both configurations.

Table S1 Adsorption distances and adsorption energies for 0-leg configuration of water molecule adsorption on graphene.		
Method	Adsorption distance (Å)	Adsorption energy (meV)
DMC ²	3.10	90
p-CCSD(T) ²		84
RPA+GWSE ²	3.05	90
CCSD(T) ³	3.06	108
BLYP-D3 ⁴	~3.08	~88
B97M-rV ⁴	~2.94	~127
DFT/CC ⁵	3.01	90

Table S2 Adsorption distances and adsorption energies for 1-leg configuration of water molecule adsorption on graphene.		
Method	Adsorption distance (Å)	Adsorption energy (meV)
DMC ²	3.46	92
p-CCSD(T) ²		76
RPA+GWSE ²	3.45	87
DMC ⁶	3.4 - 4.0	70
BLYP-D ⁶	3.47	87
BLYP-D3 ⁴	~3.29	~137
B97M-rV ⁴	~3.34	~127

DFT/CC ⁵	3.35	125
vdw-DF2 ⁷	3.42	123
vdw-DF2 ^{c09x 7}	3.42	78
optB86b-vdw ⁷	3.35	143

BLYP-D3 indeed overestimates the adsorption energy of water molecule on graphdiyne (Figure S7 and Figure S10), while the characteristic interfacial water structure is nicely described in BLYP-D3 (Figure S6, Figure S7, Figure S8), due to the presence of many-body effects. Thus, the proton transport mechanism, based on the characteristic transmembrane hydrogen bonds, is reliable in BLYP-D3.

We have made the following changes in the text:

i) Add the following sentences in the middle of page 13 in the main text.

“Due to the delicate interaction of water, none of the existing functionals is able to universally and faultlessly describe water⁴¹⁻⁴² and its self-ions⁴³⁻⁴⁵ in various realistic conditions. In particular, the water-carbon interaction is demonstrated to be extremely difficult to simulate, even high-cost Diffusion Monte Carlo (DMC), coupled cluster theory, and random phase approximation give rise to various adsorption energies of a single water molecule absorbed on graphene (see Table S1 and S2).⁴⁶⁻⁴⁹ Though dispersion-correction remedies the bad performance of DFT in describing water-carbon adsorption interactions,⁴⁶⁻⁴⁷ relatively large variations are still found in interaction energies between a single water molecule and graphene predicted by different DFT models.⁴⁹⁻⁵⁰ Despite of these discrepancies, dispersion-corrected GGA still gives rise to reasonable interfacial structures between liquid water and graphene⁵¹⁻⁵² or carbon-nanotubes,⁵³ and describes well the monolayer ice on graphite.^{54”}

ii) Add the following sentences at the top of page 14 in the main text.

“We note that BLYP-D3 exhibits an overestimate of adsorption energy of a water molecule on graphene⁴⁹ and on graphdiyne here (Figure S10b), however the characteristic interfacial water structure is well described in BLYP-D3 (Figure S6,

Figure S7, Figure S8), which is the cornerstone for proton transparency identified in the present work.”

iii) Insert Section 7 and Table S1 and S2 in supporting information to summarize the adsorption behaviors of water molecule on graphene, highlighting the comparisons to DMC and CCSD(T) results.

Reply to Reviewer 2

Comments: I am fine with the replies to my concerns, the authors performed a benchmark study that provides solid foundations for the quality of reported DFT results. For these reasons I suggest publication of the revised manuscript.

Response: We greatly appreciate the insightful comments that assisted us in improving the manuscript.